# Genetic Diversity of Dengue Vector *Aedes albopictus* Collected from South Korea, Japan, and Laos

**DOI:** 10.3390/insects14030297

**Published:** 2023-03-20

**Authors:** Jiyeong Shin, Md-Mafizur Rahman, Juil Kim, Sébastien Marcombe, Jongwoo Jung

**Affiliations:** 1Agriculture and Life Sciences Research Institute, Kangwon National University, Chuncheon 24341, Republic of Korea; 2The Division of EcoCreative, Ewha Womans University, Seoul 03760, Republic of Korea; 3Department of Biotechnology and Genetic Engineering, Faculty of Biological Science, Islamic University, Kushtia 7003, Bangladesh; 4Program of Applied Biology, Division of Bio-resource Sciences, CALS, Kangwon National University, Chuncheon 24341, Republic of Korea; 5Vector Control Consulting—South East Asia (VCC-SEA), Vientian 01000, Laos; 6Department of Science Education, Ewha Womans University, Seoul 03760, Republic of Korea

**Keywords:** *Aedes albopictus*, population structure, genetic diversity, invasive species, microsatellites, mitochondrial DNA

## Abstract

**Simple Summary:**

The Asian tiger mosquito, *Aedes albopictus,* is an invasive species that spreads disease and has recently become prevalent in its habitat due to human interference. In this study, we analyzed the genetic diversity and population structure of *Ae. albopictus* on the basis of two mitochondrial genes (cytochrome oxidase subunit 1, *COI* and NADH dehydrogenase 5, *ND5*) and sixteen polymorphic microsatellites sourced from localities in Republic of Korea (hereafter Korea), Japan, and Laos. Mitochondrial analysis showed that the genetic diversity of the Korean and Japanese populations was low, while the Laos population did not share haplotypes with the former two groups, indicating that the Korean *Ae. albopictus* were not derived from the Laos population. On the basis of microsatellite results, the Korean, Japanese, and Laos populations showed higher diversity, although there were mixed groups, which implies that the Korean cluster is distinct. Continuous international trade and overseas travel require constant monitoring of species that can transmit mosquito-borne diseases. In addition, genetic studies would provide a useful basis for understanding the genetic status of vectors and for managing vector-borne diseases.

**Abstract:**

*Aedes albopictus* is native to Southeast Asia and has emerged as a major vector for vector-borne diseases that are spreading rapidly worldwide. Recent studies have shown that *Ae. albopictus* populations have different genetic groups dependent on their thermal adaptations; however, studies on Korean populations are limited. In this study, we analyzed the genetic diversity and structure of two mitochondrial genes (*COI* and *ND5*) and sixteen microsatellites in mosquitoes inhabiting Korea, Japan, and Laos. The results indicate that the Korean population has low genetic diversity, with an independent cluster distinct from the Laos population. Mixed clusters have also been observed in the Korean population. On the basis of these findings, two hypotheses are proposed. First, certain Korean populations are native. Second, some subpopulations that descended from the metapopulation (East Asian countries) were introduced to Japan before migrating to Korea. Furthermore, we previously demonstrated that *Ae. albopictus* appears to have been imported to Korea. In conclusion, the dengue-virus-carrying mosquitoes could migrate to Korea from Southeast Asian epidemic regions, where they can survive during the severe winter months. The key findings can be used to establish an integrated pest management strategy based on population genetics for the Korean *Ae. albopictus* population.

## 1. Introduction

Invasive insect pests have recently increased in prevalence owing to climate change and increased trade, resulting in a global phenomenon [1]. Despite numerous benefits, globalization has led to the global proliferation of invasive species [2]. Invasive insects account for a considerable portion of exotic species, which not only play a role in disrupting various influences on the ecosystem but also affect biodiversity and human life [3]. Among exotic species, mosquitoes significantly impact humans and transmit diseases that are lethal to both humans and animals [4]. The spread of mosquitoes has been accelerated by increased worldwide trade in products contaminated with mosquito larvae or eggs and their introduction into new countries [5,6,7].

Mosquito-borne viruses, including dengue fever, are mostly arthropod-borne viruses transmitted to animals and humans by blood-sucking invertebrates [8]. There are 3530 mosquito species belonging to 43 genera worldwide, of which *Aedes*, *Anopheles*, and *Culex* are the 3 genera that mostly serve as carrier of mosquito-borne diseases [9,10]. Mosquitoes belonging to the *Aedes* genus are major carriers of arboviruses, such as dengue fever, yellow fever, chikungunya, and zika virus, and such diseases have recently emerged in Southeast Asian countries [9,11]. *Aedes*-mediated diseases are hazardous since various virus types occur, and there are cases where there are no vaccines or therapeutic agents available [4,11]. Mosquito-borne diseases are likely to become more prevalent because of habitat expansion under increasing globalization, more frequent international trade, and climate change [8,12]. Several mosquito species are regarded as invasive species in different nations, and several studies have been conducted to combat invasive mosquitoes [4,13]. For example, according to a study of trends in imported dengue cases in Korea and Japan, *Ae. aegypti*, which originates from Africa, has spread rapidly in places with a climate similar to that of Africa, Europe, and Southeast Asia, including Laos [14] and Japan [15], causing various fatal diseases [16]. Furthermore, *Ae. aegypti* has been studied extensively as one of the most notorious exotic species in the *Aedes* genus [17]. Additionally, various species of mosquitoes have been studied as exotic species. However, most studies have focused on specific regions, whereas genetic studies in Asian countries have been limited to only a few species [4,13,18,19,20,21].

Malaria and Japanese encephalitis frequently occur in Korea. Studies in Korea have focused primarily on mosquitoes that transmit pathogens and cause diseases, whereas there is a general lack of investigations on *Ae. albopictus* [19,20,22,23]. A genetic study of the introduction of *Ae. albopictus* was first conducted on individuals found in Jeju Island [23], whereas studies on individuals throughout Korea used only mitochondrial (mt) genes. Thus, to the best of our knowledge, there are no studies related to southeast Asian countries using mtDNA and microsatellite markers. Therefore, there is a need to study the introduction of exotic species using a diverse range of genes for improved resolution and outcomes [20]. 

The abundance of *Ae. albopictus* in Korea is expanding, owing to recent climate change [24]. Indigenous infections are often caused by imported pathogens. Since Korea has a steady inflow of imported cases, it can no longer be viewed as a virus-free country (e.g., chikungunya and dengue) [25,26,27]. However, only 11 cases of chikungunya and 194 cases of dengue fever were reported in a study of tourists who had brought the infection to Korea from abroad in 2018 [25]. As a result, the risk of disease outbreaks in Korea has been highlighted, owing to the recent rise in the number of reported cases of mosquito-borne illness. Health professionals should also closer monitor these mosquitoes for illness due to the widespread frequency and abundance of *Ae. albopictus* [24].

Genetic markers can be used to identify the origins of locations and paths of invading species from one country to another [28,29]. Molecular markers, including microsatellites and mtDNA, are extensively used in mosquito population genetics [30,31,32]. Several studies have been conducted using *COI* and *ND5* as reliable markers to identify whether a species has been imported or to assess genetic variation in a population [33]. On the basis of the mt *COI* and *ND5* sequences, we previously demonstrated the genetic diversity of two mosquito species, *Ae. albopictus* and *Ae. flavopictus*. *Ae. flavopictus* is native to the Korean Peninsula, whereas *Ae. albopictus* seems to be an introduced species [34].

*Ae. albopictus* is one of the notorious exotic species that has recently expanded its distribution areas (Global Invasive Species Database, http://www.iucngisd.org/gisd/ accessed on 29 June 2021). *Ae. albopictus* has expanded its distribution areas through human activities in Southeast Asia, which is its place of origin, and is currently found in all parts of the world except Antarctica [35,36,37]. The establishment of successful mosquito control programs can be effective when based on proper knowledge of genetic variation and population dynamics [38,39]. So far, to our knowledge, currently there is limited information about the population genetics of *Ae. albopictus* [20,23]. 

In this study, we aimed to elucidate the genetic status of *Ae. albopictus* in Korea using sixteen microsatellite markers developed in addition to the two mt markers (*COI* and *ND5*) from our recently published study [34]. Such research on genetic diversity and structure could allow effective surveillance of mosquito populations as well as help in evaluating whether an exotic species has been introduced. These findings could also be used as basic data for establishing a vector-controlling strategy for *Ae. albopictus*.

## 2. Materials and Methods

### 2.1. Sampling and DNA Extraction

Samples of 211 mosquitoes from 21 populations were captured in Korea, Japan, and Laos between 2017 and 2020 (Figure 1, Table 1). Recently, *Ae. albopictus* has been studied in the Korean population [34]. Adult mosquitoes were collected using net and BG-Sentinel traps (Biogents AG, Regensburg, Germany). Each specimen was individually maintained in tubes filled with 80% ethanol and stored at 4 °C until DNA extraction. DNA was extracted from one to three legs of each sample using the DNeasy Blood & Tissue Kit (Qiagen, Hilden, Germany).

### 2.2. Molecular Methods (Polymerase Chain Reaction [PCR])

The two regions of the mt gene (*COI* and *ND5*) were amplified using PCR using two primer pairs: albCOIF (5′-TTTCAACAAATCATAAAGATATTGG-3′) and albCOIR (5′-TAA ACTTCTGGATGACCAAAAAATCA-3′) for *COI* gene [28]; and ND5_6500F (5′-TCCTTAGAATAAAATCCCGC-3′) and ND5_7398R (5′-GTTTCTGCTTTAGTTCATTCTTC-3′) for *ND5* gene [40]. For *COI*, PCR amplifications were performed with a 25 μL reaction volume containing 0.5 μL of isolated DNA, 2.5 μL of 10 × Taq buffer, 2.0 μL of MgCl2 (25 mM), 0.7 μL of dNTP solution (2.5 mM each), 0.5 μL of each primer, and 0.3 μL of Taq DNA polymerase (Takara Bio Inc., Shiga, Japan). The PCR cycling conditions were as follows: an initial denaturation step at 95 °C for 5 min, followed by 35 cycles of denaturation at 95 °C for 30 s, annealing at 45 °C for 30 s, and elongation at 72 °C for 45 s, and a final extension at 72 °C for 7 min. For *ND5*, PCR mixture was the same as *COI*. The amplification conditions were as follows: initial denaturation at 98 °C for 5 min, followed by 10 cycles at 95 °C for 1 min, 45 °C for 1 min, 72 °C for 1 min 30 s, 30 cycles at 95 °C for 1 min, 46 °C for 1 min, 72 °C for 1 min 30 s, and a final extension at 72 °C for 3 min. PCR products were separated using 2% agarose gel electrophoresis (Sigma-Aldrich, Saint Louis, MO, USA) and sequenced by Cosmo Genetech (Seoul, Korea) using an ABI 3730xl DNA Analyzer (Applied Biosystems, Foster City, CA, USA).

Sixteen polymorphic microsatellite primers were used in this study [41,42,43] (Appendix A). The forward sequence of each primer was labeled with four different fluorescent dyes (FAM, NED, VIC, and PET), and pig tail (GTTTCTT) was attached to reverse primers. PCRs were performed in a total volume of 16 μL consisting of 8 μL of 2X QIAGEN Multiplex PCR Master Mix (Qiagen), 6.58 μL of sterile water, 0.16 μL forward primer, 0.8 μL pig-tailed reverse primer, 0.3 μL of genomic DNA, and 0.16 μL of each fluorescent dye (FAM, NED, VIC, and PET). 

For PCR, samples were denatured at 95 °C for 15 min, followed by 90 s of specific annealing step (14 cycles at 63 °C, 7 cycles at 58 °C, and 20 cycles at 55 °C), elongation step at 72 °C for 30 s, and a final extension step at 72 °C for 20 min. Genotyping was performed using an ABI 3730xl DNA Analyzer (Applied Biosystems, Foster City, CA, USA), and the data were analyzed using GeneMapper software v.5 (Applied Biosystems, Foster City, CA, USA).

### 2.3. Mitochondrial DNA Data Analysis for Genetic Diversity and Gene Flow

The sequences of two mt genes (*COI* and *ND5*) were aligned using Clustal W [44] plugin on Geneious Prime 2022.2.2 (https://www.geneious.com accessed on 29 June 2021; Biomatters, New Zealand). Concatenated sequences were derived from two mt genes using gene flow and genetic diversity analyses. The number of haplotypes, number of segregating sites (*S*), haplotype diversity (Hd), nucleotide diversity (π), and average number of nucleotide differences (k) were analyzed for genetic diversity using DnaSP.6.12 [45]. The values of the pairwise fixation index (*F*_ST_) were calculated to investigate genetic differentiation among populations using Arlequin v.3.5 [46]. Additionally, principal coordinate analysis (PCoA) was performed using GenAlEx v. 6.51b2 [47] on the basis of pairwise *F*_ST_ values. To determine the population structure, analyses of molecular variance (AMOVA) were conducted using Arlequin v.3.5 [46] with the option of locus-by-locus and 1000 permutations. Each population was divided into three independent ‘groups’ by nation and analyzed on ‘within-population’, ‘among-population within group’, and ‘among-group’ bases: Group 1 contained samples from Korea; Group 2 contained samples from Japan; and Group 3 contained samples from Laos. In addition, AMOVA analysis was conducted by dividing Korea into east–west, north–south, and five provinces (Gangwon-do, Gyeonggi-do, Gyeongsang-do, Jeolla-do, and Chungcheong-do) (see Appendix A).

To investigate population expansion, the selective neutrality test was evaluated by Tajima’s *D* [48] and Fu’s *F*s [49] using Arlequin v.3.5 [46]. The mismatch distribution was estimated using DnaSP v.6 [45].

A genealogical relationship among the haplotypes was established using the minimum spanning network algorithm, as implemented in Population Analysis with Reticulate Trees, PopART v.1.7 [50]. The minimum spanning network was confirmed using the *CO1* and *ND5* sequences from NCBI [14,20,40,51,52,53] to determine the genealogical relationship with more closely related countries using PopART v.1.7 [50].

### 2.4. Microsatellite Data Analyses

Sixteen previously published primer sets [41,42,43] were used for microsatellite genotype data analyses. Microsatellite primers were subjected to BLAST at VectorBase to confirm genomic similarity for reliability (Appendix A). The microsatellite datasets created a data format using GenAlEx v.6.51b2 [47] converted into Arlequin, GENEPOP, and FSTAT typefiles using PGDSpider version 2.1.1.5 [54] to calculate population genetics and allelic diversity calculations. 

The genetic diversity of the 16 loci was characterized by estimating the mean number of alleles (*N*_a_), allelic range (*A*_R_), observed heterozygosity (*H*_o_), expected heterozygosity (*H*_E_), and Hardy–Weinberg equilibrium (HWE) using Arlequin v.3.5 [46]. *P*-values were adjusted using Bonferroni correction in R [55]. GENEPOP v.4.7.5 on the web (https://genepop.curtin.edu.au accessed on 29 June 2021) was used to assess for linkage disequilibrium (LD) between pairs of loci using probability tests on 1000 dememorizations with 100 batches (1000 iterations per batch). Arlequin v.3.5 [46] was used to calculate pairwise *F*_ST_ and *R*_ST_ with 1000 permutations, and the AMOVA was employed with locus-by-locus and 1000 permutations to determine the genetic differentiation among populations. FSTAT 2.9. [56] was used to determine the inbreeding coefficient (F*_IS_*) and allelic richness (r) in each population. GenAlEx was used to generate visual data to facilitate an easier comparison of the mean allelic patterns of the entire population [47].

The Wilcoxon signed-rank tests of heterozygosity were conducted using Bottleneck [57] to search for recent population changes. Calculations were performed using three different models: the stepwise mutation model (SMM), infinite alleles model (IAM), and a two-phase model (TPM) using 30% of indels larger than one repeat with 1000 replications. The demographic history of populations was investigated using the Garza–Williamson index (M-ratio), and recent bottleneck events were assessed using Arlequin v.3.5 [46]. 

Bayesian clustering analysis was performed to determine population genetic structure using STRUCTURE v.2.3.4 [58] making use of an admixture model with correlated allele frequencies among populations. The number of populations, *K*, ranged from 1–8, executing 10 independent chains with 100,000 burn-in steps and 500,000 iterations. The most likely number of *K* was determined using STRUCTURE HARVESTER [59]. The results were visualized using CLUMPAK [60].

## 3. Results

### 3.1. Genetic Diversity and Population Structure of Mitochondrial DNA

Two nucleotide sequences, 658 bp *CO1* and 423 bp *ND5*, were obtained from 211 mosquito samples, including 148 Korean samples from our previous study [34], 35 samples from Japan, and 28 samples from Laos (Table 1). Thirty-eight haplotypes were observed in the mtDNA sequence analysis (Appendix A). The overall haplotype and nucleotide diversity were 0.678 and 0.00132, respectively. The average number of nucleotide diversities (*k*) was 1.426 overall. The Vientiane, Laos (VL), population had the most significant number of haplotypes, with 11 haplotypes from 28 individuals, and 9 individuals had a single haplotype. The Yangsan (YS) population had the highest haplotype diversity (0.758), followed by VL population (0.706) (Table 1).

Pairwise genetic distance analysis (*F*_ST_) of the tested specimens showed that Geoje, Korea (GJ), population showed the highest level, followed by VL and TJ (Tokyo, Japan) populations that distinguished themselves from the other populations (Appendix A). Principle component analysis (PCoA) based on pairwise *F*_ST_ also demonstrated that GJ, VL, and TJ were most distantly related to the other populations, and that most of the populations in Korea were closely related (Figure 2).

The genetic differences in the populations were analyzed by molecular variance (AMOVA) using Arlequin software. The AMOVA was used to compare genetic variations within and between the populations. In this study, the AMOVA was performed by grouping each country (Korea, Japan, and Laos). The highest genetic variation (56.02%) was observed among groups. Within populations, however, a relatively low variation (27.82%) was observed. The least variation (16.15%) was found among populations within groups (Table 2). The AMOVA results, conducted by dividing Korea into east–west, north–south, and provincial units, found no significant differences between groups (Appendix A).

Neutral tests (Tajima’s *D* and Fu’s) were employed to analyze the signatures of historical demographic events. The goal of the Tajima’s test is to identify sequences that do not fit the neutral theory model when mutation and genetic drift are in equilibrium. Fu’s *F*s is built upon the Ewens’ sampling distribution based on haplotype (gene) frequency distribution, which is different from the Tajima’s *D*. Tajima’s *D* and Fu’s *F*s neutrality tests showed that the VL population had a statistically significant negative value (−1.94835, *p* = 0.01), indicating that the VL population has an excess of alleles (Table 1). The group with significantly negative values in both tests was Jeonju (JJ) population. Tests of selective neutrality in other populations showed non-significant values for Tajima’s *D* (*p* > 0.05) and Fu’s *F*s (*p* > 0.05) in most geographic groups. All populations under study had unimodal mismatch distributions, which supported the standard neutral model of a panmictic population of constant size (Figure 3). Whereas the Japanese and Laos populations exhibited a unimodal shape, and the Korean populations exhibited an L-shape, the population of Laos had a larger unimodal form than that from Japan. However, a negative Tajima’s *D* denotes a surplus of lower-frequency polymorphisms compared with what is expected by the neutral theory model. When Fu’s *F*s is negative, more alleles are present than would be expected given the current level of population expansion.

The Laos (VL) and Japan (TJ) populations formed lone haplotypes in their haplotype networks (Figure 4). Japanese (TJ) populations shared haplotypes (denoted as H) with Korean (GJ) populations. Korean (GJ) populations (H17 and H18) may have originated from the Japanese population (H-21). The Korean Gyeongju, GY (H-19) populations were connected to the Laos population (Figure 4). It is likely that Japanese mosquito populations were first introduced on the southeastern coast (GJ) of the Korean peninsula. The Korean population demonstrated a star-like shape extending from one focal haplotype (H1), which was present in 115 out of 148 entities with the highest frequency. Some local entities were closely related to Japan (TJ) and Laos (VL) populations. Among TJ populations, 29 out of 35 entities belonged to H21, including Jeungdo (JD) population of Korea. The number of haplotypes was relatively low compared with the number of entities. Laos populations did not share haplotypes with other populations and had a greater number of haplotypes compared with the number of entities. The haplotype network using NCBI-acquired *CO1* sequences was clearly distinguished into three clades (Korean, Southeast Asian, and Japanese clades) (Appendix A). This result also shows that Korea’s GJ population forms the same clade as that of Japan and other countries. However, in the *ND5* sequence-based haplotype network, the clades could not be distinguished as in the *COI* sequence-based network (Appendix A).

### 3.2. Genetic Diversity and Population Structure of Ae. albopictus Based on Microsatellite Markers

Based on the partial mt sequence analysis (658 bp *CO1* and 423 bp *ND5*), 131 out of the 211 entities, which corresponded to 19 populations, were selected and analyzed using 16 microsatellites (Appendix A, Figure 5). The diversity of the various groups was relatively high in the mean allelic patterns produced by GenAlEx (Figure 5). In total, 281 alleles were identified in 16 different loci using GenAlex. The population of Laos had the highest *N*_A_, followed by those in Tokyo, Japan, and Jeonju, Korea. Allelic size of 16 loci ranged from 1 to 28. In all 16 loci, the ranges of the mean values of *H*_o_ and *H*_E_ were 0.117–0.545 and 0.403–0.628, respectively. The Cheongyang, Korea (CY), population had the highest *H*_o_ (0.545), while JD population had the lowest *H_o_* (0.117). In most populations, *H*_o_ was lower than *H*_E_, indicating an excess of homozygote genotypes at most loci. In addition, most populations deviated considerably from HWE after Bonferroni correction. The inbreeding coefficient (*F*_IS_) ranged from 0.385 to 0.828, whereby a high *F*_IS_ value indicates a high level of inbreeding in the population, while a low *F*_IS_ value indicates a low level of inbreeding. The allelic richness (r) ranged from 1.846 (Yeoncheon, Korea) to 2.471 (JJ; Jeonju, Korea). Allelic richness (r) of microsatellites was high in indigenous Korean populations; however, there was no significant difference between the Laos and Korean populations. The linkage disequilibrium test assessed the independence of the 16 loci, which showed no significant results in all samples.

STRUCTURE analysis can reveal historical population structures and a mix of populations in the past by reconstructing the ancestry of individuals in the population and determining the contributions of various ancestral populations (Figure 6). Two methods, Bayesian analysis and Evanno’s method, were used to evaluate the hypothetical ancestral subpopulations and *K* values (Appendix A). The Evanno delta K (Δ*K*) result supported the presence of two clusters for the nineteen populations, suggesting that the population can be divided into two clusters/subpopulations. However, the two clusters did not differ significantly from one another. At the highest value (*K* = 4), eight Korean populations (WJ-II, YE, SE, GY, YJ, GC, CY, and SC) had distinct genetic groups (orange color), whereas the other regions of Korea (DJ, GJ, JD, CC, JJ, and WJ-I) shared genetic clusters with TJ (Japan) and VL (Laos).

Pairwise *F*_ST_ and *R*_ST_ to assess the genetic distance between different populations were 0.117~0.438 and −0.028~0.581, respectively, which were generally lower for the mt data. The pairwise *F*_ST_ and *R*_ST_ values were slightly different; however, the GJ population of Korea and the VL population had a greater distance compared with the other groups (Appendix A). An AMOVA based on pairwise *F*_ST_ was conducted to determine the cause of the genetic variations observed within the studied populations (Table 3). The variation among groups in each country was low. However, the variations among populations within groups and within populations were 23.77% and 79.01%, respectively, suggesting that most of the variations were within populations. 

The Wilcoxon test (two-tailed Wilcoxon signed rank test for heterozygosity excess) and Garza–Williamson index (M-ratio) test were conducted to assess the recent bottleneck phenomenon, which showed that most populations suffered from a genetic bottleneck in the past (Appendix A). The Garza–Williamson index (M-ratio) showed significant results. An M-ratio close to 1 indicates that the population was in a fixed state, whereas a lower value indicates that there was a bottleneck in the past (with a critical value of 0.68, indicating a bottleneck). The Wilcoxon test can show whether significant heterozygote excesses experienced a bottleneck phenomenon. Although significant results were observed in most populations, the Laos population did not show significant results. 

## 4. Discussion

Climate change has created suitable climatic conditions, especially in terms of temperature changes, for many insects to adapt to new habitats [61]. Insect life cycles, population densities, genetic shifts, and broad temperature adaptability may be affected by climate change [62,63]. Additionally, nondiapausing eggs have improved cold exposure endurance compared with eggs from tropical populations, demonstrating that cold tolerance comes at a fitness cost in terms of egg viability [63]. Although research on the introduction and spread of disease-controlling vectors is ongoing around the world, Korea has yet to conduct any such studies [20,39,64].

Dengue outbreaks occurred in Japan in 2014 [15], Laos in 2018 [65], and the first imported patient in Korea in 1995 (to date, no outbreak has been reported in Korea; the first case of dengue fever was a woman who had traveled to Sri Lanka) [27,66]. In this study, two different types of genetic markers (mtDNA and microsatellite markers) have been used in the population genetic analysis of *Ae. albopictus*. The *COI* and *ND5* sequences were used to identify the three clades in Figure 4: Clade I (Laos, blue-shaded), Clade II (Japan with some population of Korea, red-shaded), and Clade III (only Korea, yellow-shaded) were clearly distinct from one another in the minimum spanning tree of *COI* sequences with NCBI obtained (Appendix A). The minimum spanning tree of *COI* sequences with NCBI acquired (Appendix A) also clearly distinguished three clades: Clade I (Southeast Asia, blue-shaded), Clade II (Japan with others, red-shaded), and Clade III (Korea, yellow-shaded). The only *COI* sequences with NCBI in this study showed similar three clades (Figure 4). However, two Korean haplotypes, H13 and -14 (Clade III) were shared with Japanese haplotypes H17 (Clade II) (Appendix A). It is likely that Japanese mosquito populations were first introduced on the southeastern coast (GJ) of the Korean peninsula. In addition, a star-like haplotype of Korean haplotype, H1 indicated that ancestral haplotype was centered on it. This star-like pattern indicates the population expansion; most haplotypes originated recently. However, this genetic structure was notably different from that of other populations in Laos and Japan (Figure 4).

The Korean population of the *Ae. albopictus* has a distinct genetic structure that was not observed in the mt (*COI* and *ND5*) analyses. However, this genetic structure was not notably different from that of other populations in Laos and Japan (Appendix A, Figure 6). The results of similar research on the genetic diversity and gene flow of Laos populations indicated that there was a slight variation within the mtDNA [14]. Mt analysis showed that the Japanese and Korean populations showed overall low genetic diversity, except in certain populations with marginal genetic structures. However, captured sample sizes were insufficient in some localities in Korea, Laos, and Japan. Hence, we were unable to provide a finer resolution or clearer clarification of genetic diversity. In contrast, the Laos population showed high genetic diversity (Table 1). The *Wolbachia* strain, however, may shorten the lifetime of adult *Ae. aegypti* [67]. There are possible explanations for the lower genetic diversity of *Ae. albopictus* in Korea. *Wolbachia* may reduce mt variation in *Ae. albopictus* populations [68,69], especially since more than 99% of the Korean *Ae. albopictus* were infected with the harbored *Wolbachia* [70]. Further, *Wolbachia* infects mainly arthropod species, such as mosquitoes, and can cause low mt diversity [68,71,72,73].

*Ae. albopictus* is an invasive species that has recently spread from Southeast Asia [35,74]. Foreign trade has grown as globalization has accelerated, and larvae and eggs mixed with trade products, such as used tires and bamboo, has assisted in its spread [5,6]. Previous studies have shown that the diversity is highest in Southeast Asia, where these mosquitoes originate, while the diversity is lower in the other regions [14,52,75]. In this study, a low level of genetic diversity was observed in the Korean and Japanese populations, which is consistent with the results of previous studies [34,76]. The Laos population was shown to have a high level of genetic diversity, which is thought to be the origin of the mosquitoes, which also supports the findings of previous studies [14,20,23,76]. Owing to the high humidity and temperatures of Southeast Asia, including Laos, which is assumed to be the origin of *Ae. albopictus*, the environment is favorable for mosquito growth [77]. In contrast, in Korea, the weather is dry with cold winters with average temperature of 10 °C, reducing the number of mosquitoes and leading to low genetic diversity [20,78]. The patterns observed in the Korean populations (low genetic diversity, insignificant mismatch distribution, and one focal haplotype shared by various populations) suggest that the effective population size could be decreased by humans and changes in the natural environment [52,75,79]. However, the Laos populations showed genetic structuring similar to those of certain Korean populations but did not share haplotypes (Figure 4), suggesting that some of the *Ae. albopictus* in Korea do not originate from Laos or Tokyo [20,23]. In addition, previous studies have shown that clusters of *Ae. albopictus* were divided according to temperate and tropical regions of Asia [14,39,80,81], and the Korean populations had distinct haplotypes while sharing certain haplotypes with other populations from different countries. Therefore, an in-depth study of the origin of *Ae. albopictus* is required in Korea [20]. In a study conducted in Japan, the Tokyo population also had low mt genetic diversity; however, there was no evidence of multiple origins [76]. Rather, it was observed that the North American populations originated in Japan [80,82].

Previous studies have demonstrated that a combined analysis of microsatellite and mt data leads to more information since mt genetic markers reflect only the history of maternal inheritance in the group; they are insufficient for finding the overall biological record and do not follow neutral evolution based on various factors [83,84,85]. Therefore, microsatellites are widely used to evaluate the genetic variation and structure of mosquitos because of their high mutation rates, codominant expression, and universal distribution [86].

In this study, microsatellite analysis revealed a unique cluster in the Korean populations and a mixed cluster with other countries. First, Korean populations had similar genetic diversity to Laos populations: *H*_o_ was 0.146, −0.117–0.545, and 0.211 in Laos, Korean, and Japanese populations, respectively, which were similar except in certain populations. There were no significant differences in *N*_a_ (number of alleles; the changes in the number of alleles commonly found in several loci within a population indicate the number of alleles present in the population) or *A*_R_ (allelic range; refers to the difference between minimum and maximum repetition in microsatellite data) (Appendix A, Figure 5). This finding is consistent with the results of previous studies, and it was observed that there was no significant difference in genetic diversity between countries of origin and the affected countries [87]. The Korean, Japanese, and Laos populations showed genetic characteristics of the original population, such as a high number of alleles coupled with a high number of private alleles occurring at high frequencies [87,88].

Second, genetic structure was observed in the Korean population, which showed that the Korean populations were concentrated on one focal haplotype (Figure 4). However, the microsatellite analysis showed that the Korean populations formed a genetic structure and that certain populations had the same genetic pattern as foreign populations (Figure 6). The AMOVA showed a high level of variation among the populations (Table 3). With the rapid expansion of Asian tiger mosquitoes, in-depth research has been conducted on the origin of these mosquitoes. Several studies have shown that only some populations in Asia spread around the world; however, microsatellite information on Korean populations was not included in those that spread to other countries [64,80,87]. The unique cluster of the Korean population in our study is thought to be the original *Ae. albopictus* population in Korea. However, mixed clusters were also observed, which is consistent with the previous studies showing that Japan and China were considered the origin of *Ae. albopictus* [64,82,87]. Other studies have also supported the idea that Asia, which can be divided into continental Asia, Japan, and Korea, is the origin of *Ae. albopictus* [39,80]. Korean populations have characteristics (i.e., cold tolerance and fitness (egg viability in cold temperature) and body sizes measured by wings) of the original population, suggesting that there is a high possibility that Korea may be the area of origin of *Ae. albopictus* [20,39,64,87]. Mixed results are observed in some populations. Multiple introductions can cause a certain level of mixing in the early stages of invasion, which results in increased genetic diversity that is favorable for expansion and adaption [89,90]. A previous *Ae. aegypti* study also showed mixed results, suggesting that the Geoje population in Korea, in particular, may have been invaded by multiple introductions [29]. Other studies of *Ae. albopictus* populations have shown that admixture is confirmed across Asian countries [87]. This may have resulted from the increased international trade and travel [23,91].

The evaluation of fewer mosquito populations with inadequate expression resolution was a limitation of this study. The introduction of the Korea (southeastern coast (GJ) of the Korean peninsula) *Ae. albopictus* mosquito is further supported by genetic diversity, variations in haplotypes and shared haplotypes with Japan (Figure 4), network, and STRUCTURE analysis, as well as by previous studies [29,34]. Therefore, it is necessary to perform more comprehensive data analyses (i.e., the microsatellite genes throughout the amplicons of the sanger sequencing) on a number of mosquito samples. This would significantly improve the data results as well as provide clarification when drawing a conclusion from the study.

Every year, Korea suffers from persistent outbreaks of mosquito-borne diseases mediated by the *Anopheles* and *Culex* genera, such as malaria and Japanese encephalitis [19,22,92]. Therefore, Korean mosquito research has prioritized the latter genera and has paid little attention to the *Aedes* genus [20,34,93,94]. However, Korea is facilitating the spread of potential vector mosquitoes owing to climate-change-induced increases in temperature and moist environmental composition [78,95,96]. In addition, there is a constant influx of patients from abroad with mosquito-borne diseases, such as zika virus and dengue fever, carried by *Ae. albopictus* [25,26]. A previous study confirmed that *Ae. albopictus* inhabiting Korea can transmit the zika virus under laboratory conditions [97]. To date, there have been no cases of mosquito-borne diseases being transmitted by infected patients, but since the Korea *Ae. albopictus* has sufficient capacity to transmit and establish these diseases, a basic vector control strategy is needed [25,26].

## 5. Conclusions

Genetic diversity and population structure of *Ae. albopictus* were analyzed mainly in Korea, including samples from two neighboring countries. The results confirmed that the Korean *Ae. albopictus* population had low genetic diversity but had distinct clusters from overseas populations and mixed clusters. It was observed that there was the possibility of introduced populations due to globalization and increased international trade and travel. Increased temperature due to climate change, domestic patients, and increased population density present a risk of mosquito-borne diseases in Korea. In addition, the increasing temperature and humidity in Korea make it a suitable environment to which mosquitoes can adapt, making it more likely that mosquito-borne diseases will spread. The results of this preliminary investigation of the potential vector may be used to better understand the geographic range of *Ae. albopictus* species, population size, and monitoring, which may have an impact on human arbovirus transmission and may aid in the development of a national-level policy control strategy.

## Figures and Tables

**Figure 1 insects-14-00297-f001:**
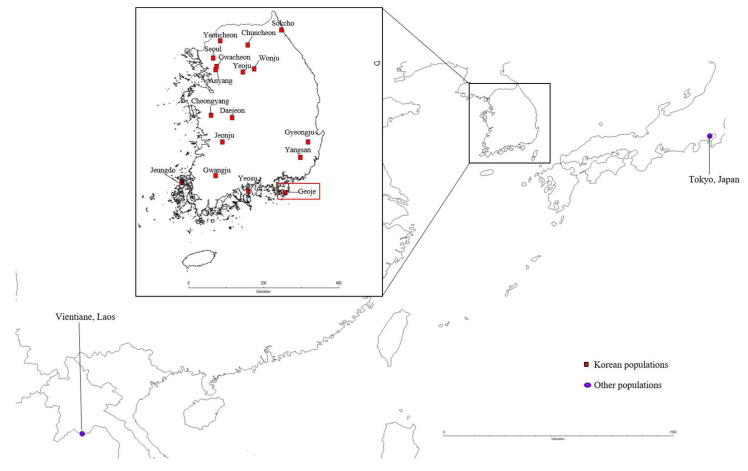
Sampling sites of *Aedes albopictus* in Korea, Laos, and Japan. Korean sampling locations (*n* = 17) are marked by red squares, whereas Japan (*n* = 1) and Laos (*n* = 1) sample localities are marked by violet circles. A distribution map was created using DIVA-GIS (version 7.5, www.diva-gis.org accessed on 29 June 2021) on the basis of the locations’ geographic coordinates. Detailed sampling information is provided in Table 1.

**Figure 2 insects-14-00297-f002:**
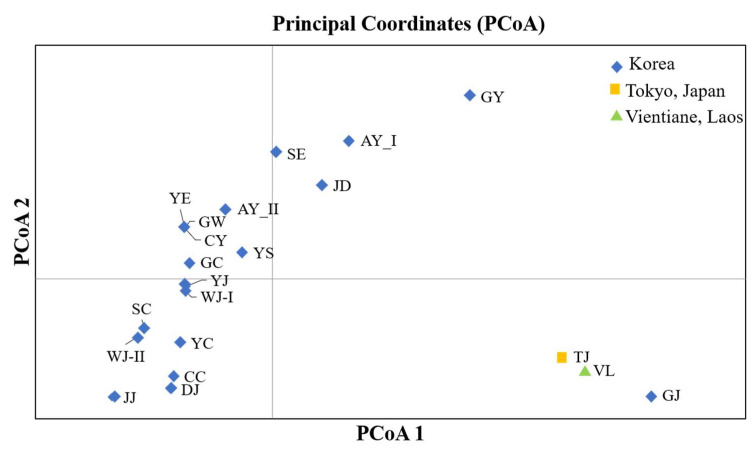
Principal Coordinates Analysis (PCoA) generated using mitochondrial DNA (mtDNA) of *Aedes albopictus.* The analyzed mtDNA sample sequences were collected from Korea (blue), Tokyo, Japan (yellow), and Vientiene, Laos (green).

**Figure 3 insects-14-00297-f003:**
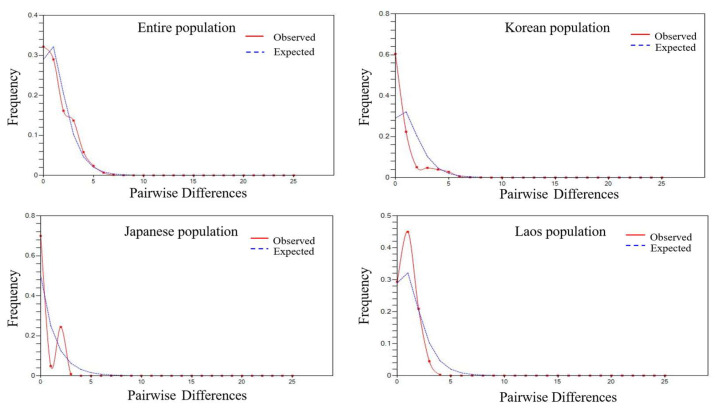
The observed pairwise mismatch distribution of mitochondrial DNA in *Aedes albopictus* as compared with the expected distribution based upon a model of population expansion. The model of population expansion was calculated using DnaSP software.

**Figure 4 insects-14-00297-f004:**
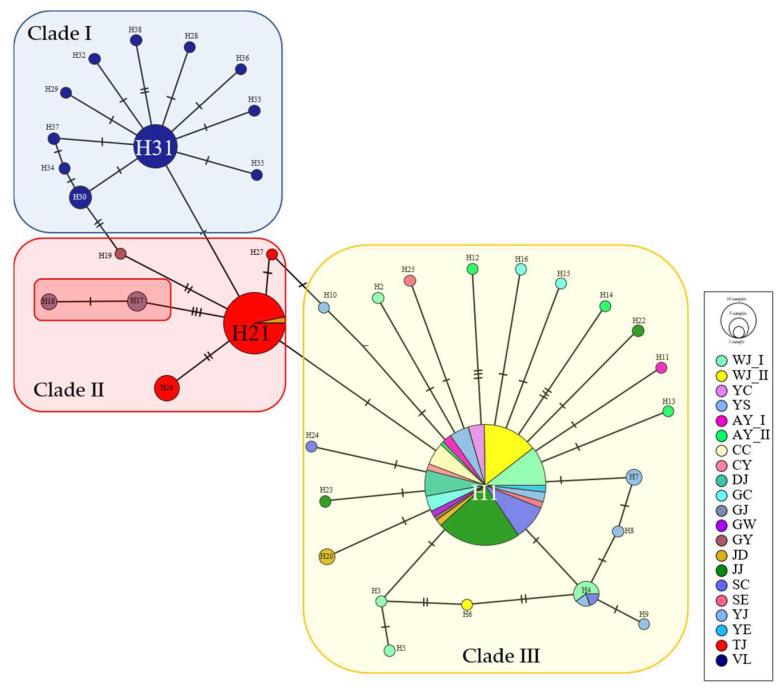
Minimum spanning network based on the combined *CO1* and *ND5* sequences of *Ae. albopictus.* The minimum spanning network algorithm as implemented in Population Analysis with Reticulate Trees, PopART, a web-based software package. The red-boxed Korean haplotypes (H17 and H18) were well connected to the Japanese clade (H21). Most of the Korean haplotypes are grouped, including H1 (yellow-boxed), Japan haplotypes are grouped, including H21 (red-boxed), and all Laos haplotypes are grouped, including H31 (blue-boxed). A detailed description is provided in Appendix A.

**Figure 5 insects-14-00297-f005:**
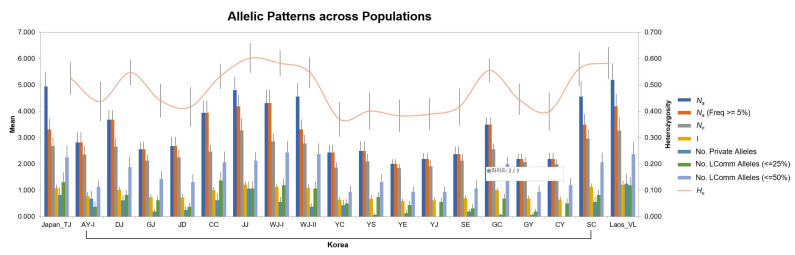
Mean allelic patterns across 19 populations of *Aedes albopictus. N*_a_ = No. of Different Alleles; *N*_a_ (Freq ≥ 5%) = No. of Different Alleles with a Frequency ≥ 5%; *N*_e_ = No. of Effective Alleles = 1/(Sum pi^2^); I = Shannon’s Information Index = −1* Sum (pi * Ln (pi)); No. Private Alleles = No. of Alleles Unique to a Single Population; No. LComm Alleles (≤25%) = No. of Locally Common Alleles (Freq. ≥ 5%) Found in 25% or Fewer Populations; No. LComm Alleles (≤50%) = No. of Locally Common Alleles (Freq. ≥ 5%) Found in 50% or Fewer Populations; Yellow line is *H*_e_ = Expected Heterozygosity = 1 − Sum pi^2^.

**Figure 6 insects-14-00297-f006:**
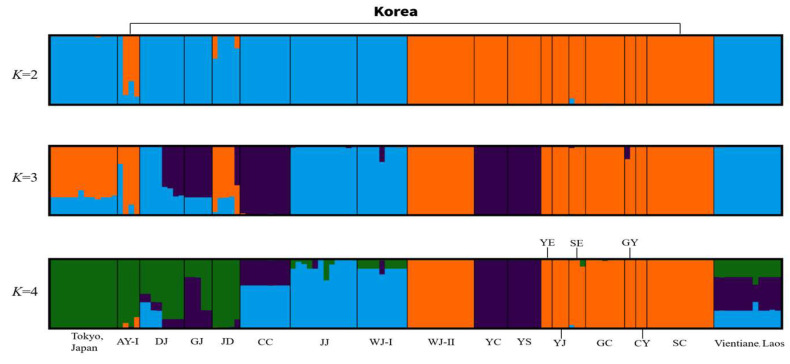
The bar plots of population structure generated by STRUCTURE software at K = 2–4. Different geographic locations are shown in STRUCTURE based on 16 microsatellite loci. The four distinct colors (orange, green, violet/purple, and sky blue) of each vertical solid bar denoting an individual/single population are also represented.

**Table 1 insects-14-00297-t001:** Geographic information, haplotypes, and genetic analysis of the concatenated sequences of the mitochondrial DNA (*CO1*, *ND5*) from 21 different genetic populations of *Aedes albopictus*.

Locality	Sample ID	Geographic Coordinates	Date	Sample Size	No. Haplotypes	*S*	k	Hd	π	Tajima’s *D*	Fu’s *F*s
Wonju (2017)	WJ-I	37°22′52.6″ N 127°53′36.1″ E	2017.07	18	5	4	0.725	0.549	0.00067	−1.12822	−2.0958
Wonju (2020)	WJ-II	37°22′52.6″ N 127°53′36.1″ E	2017.07	18	2	3	0.333	0.111	0.00031	**−1.71304 (*p* = 0.022)**	0.65061
Yeoncheon	YC	37°58′35.6″ N 127°04′05.3″ E	2017.07	5	1	0	0	0	0	0	0
Yangsan	YS	35°31′13.9″ N 129°00′40.1″ E	2017.07	12	6	4	1.152	0.758	0.00107	−0.45947	**−2.89747**
Anyang (2018)	AY-I	37°22′01.7″ N 126°57′39.0″ E	2018.06	4	2	1	0.5	0.5	0.00046	−0.61237	0.17185
Anyang (2020)	AY-II	37°22′01.7″ N 126°57′39.0"E	2020.08	4	4	6	3.167	1	0.00293	−0.31446	−1.15708
Chuncheon	CC	37°53′15.0″ N 127°44′02.9″ E	2017.08	7	1	0	0	0	0	0	0
Cheongyang	CY	36°24′28.5″ N 126°51′04.8″ E	2017.09	2	1	0	0	0	0	0	0
Daejeon	DJ	36°21′02.3″ N 127°21′35.5″ E	2017.09	8	1	0	0	0	0	0	0
Gwacheon	GC	37°26′00.0″ N 126°59′00.1″ E	2020.07	7	3	2	0.571	0.524	0.00053	−1.23716	−0.9218
Geoje	GJ	34°46′43.8″ N 128°38′42.1″ E	2017.07	5	2	1	0.6	0.6	0.00056	1.22474	0.62615
Gwangju	GW	35°07′53.0″ N 126°57′30.0″ E	2017.09	2	1	0	0	0	0	0	0
Gyeongju	GY	35°50′41.4″ N 129°11′43.4″ E	2017.07	2	2	3	3	1	0.00278	0	1.09861
Jeungdo	JD	34°59′15.8″ N 126°08′07.3"E	2017.07	5	3	2	1	0.8	0.00093	0.24314	−0.47542
Jeonju	JJ	35°50′37.2″ N 127°07′09.0″ E	2017.08	28	3	2	0.143	0.14	0.00013	**−1.5106 (*p* = 0.045)**	**−2.26798**
Sokcho	SC	38°12′10.5″ N 128°33′07.5″ E	2020.09	13	3	2	0.308	0.295	0.00028	−1.46801	**−1.4015**
Seoul	SE	37°36′32.9″ N 126°54′06.5″ E	2020.07	3	2	1	0.667	0.667	0.00062	0	0.20067
Yeoju	YJ	37°18′43.5″ N 127°36′52.8″ E	2020.07	3	1	0	0	0	0	0	0
Yeosu	YE	34°47′57.0″ N 127°44′52.4″ E	2020.06	2	1	0	0	0	0	0	0
Tokyo, Japan	TJ	35°33′01.2″ N 139°44′16.5″ E	2018.09	35	3	3	0.561	0.301	0.00052	−0.51759	0.39764
Vientiane, Laos	VL	17°57′42.6″ N 102°36′30.2″ E	2018.07	28	11	10	1.013	0.706	0.00094	**−1.94835 (*p* = 0.010)**	**−8.85951**
Total/average				211	38	41	1.426	0.678	0.00132	−0.44959	−0.80624

Significant values (*p* < 0.05) of the neutrality test (Tajima’s *D* and Fu’s *F*s) are shown in boldface after the Bonferroni correction. No. haplotypes, number of haplotypes; *S*, number of segregating sites; *k*, average number of nucleotide differences; Hd, haplotype diversity; π, nucleotide diversity.

**Table 2 insects-14-00297-t002:** Analysis of Molecular Variance of *Aedes albopictus* with concatenated sequences of mitochondrial DNA.

Source of Variation	df	Sum of Squares	Variance Components	Percentage of Variation (%)	F-Index *
Among groups	2	67.077	0.58191	56.02	0.56024
Among populations within groups	18	27.71	0.16776	16.15	**0.36729**
Within populations.	190	54.91	0.28900	27.82	**0.72176**
Total	210	149.697	1.03867		

df, degrees of freedom. * Significant values (*p* < 0.05) of Fixation Indices are shown in boldface after the Bonferroni correction.

**Table 3 insects-14-00297-t003:** Hierarchical analysis of molecular variance (AMOVA) for *Aedes albopictus* based on 16 microsatellite loci.

Source of Variation	df	Sum of Squares	Variance Components	Percentage of Variation (%)	F-Index
Among groups	2	58.357	−0.15670	−2.78	−0.02780
Among populations within groups	16	336.027	1.33962	23.77	**0.23124**
Within populations	243	1082.238	4.45365	79.01	**0.20987**
Total	261	1476.622			

df: degrees of freedom. Significant values (*p* < 0.05) of Fixation Indices are shown in with boldface after the Bonferroni correction.

## Data Availability

All data produced are available in this manuscript. The accession numbers (MW526509–MW526719) of cytochrome oxidase subunit 1, *COI*, and NADH dehydrogenase 5, *ND5* (MW526720–MW526930) of NADH dehydrogenase 5, *ND5* were deposited to the NCBI GenBank.

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
