# Peer review of "Genetic Diversity of Dengue Vector Aedes albopictus Collected from South Korea, Japan, and Laos"

_insects, 2023, doi:10.3390/insects14030297_

Round 1

Reviewer 1 Report

Understanding vector genetic diversity could inform risk assessment for emerging vector-borne disease outbreaks. This study explores variation of COI, ND5, and microsatellites in Ae. albopictus mosquitoes. The authors used modern population genetics metrics to characterize markers from each population. They used quite a few different types of analyses, which makes this work particularly informative compared to similar studies. Unfortunately, they neglected to explain the purpose of each type of analysis as it was introduced.

This work is organized in the traditional Results and Discussion sections. One suggestion would be to reorganize it to a combined ‘Results and Discussion’ sections, which would allow the authors to expand their explanations of each analytical method as the ms progresses.

The availability of a complete genome for this species (Palatini U, et al. Improved reference genome of the arboviral vector Aedes albopictus. Genome Biol. 2020 Aug 26;21(1):215. See also vectorbase.org) makes the use of microsatellites somewhat outdated. Understanding that short read similarity searches have limited utility, I recommend that the authors BLAST all the primers used to determine which genes were likely represented in their analyses. See https://vectorbase.org/vectorbase/app/workspace/blast/new

Ln 60- The authors refer to ‘fatal’ diseases several times in the Introduction. The disease burden caused by arboviruses is much greater than case fatality rates per se. I recommend removing the word ‘fatal’ and replacing it with a word or phrase that more accurately summarizes the impact of these arboviruses on human health.

Ln 76- Mosquitoes don’t ‘cause’ disease. Rather, mosquitoes transmit pathogens (the pathogens cause disease..)

Ln 83- This sentence is unclear- please clarify- In foreign cases, imported cases have led to indigenous infection.

Ln 85- This sentence is unclear- please clarify- However, investigations of travelers having introduced the infection 85 from overseas have continuously included cases of dengue fever 194 and chikungunya 11 86 cases in 2018 only.

Ln 99- The authors state that their goal is to “to determine the introduction of Ae. albopictus in Korea”. Did the reported analyses satisfy that goal?

Ln 224- The authors state that population VTE has the largest number of haplotypes. However, this evidence is not clear in either Table 1 or Table S2. Please clarify….

Ln 226- Please define all acronyms upon first use, eg., VL and TJ.

Table 2 has a typo.

Table 2. Some readers may not understand the differences between ‘population and group-wise comparisons. Please add a sentence to the Results section, explaining the differences between the metrics.

Starting in Ln 235- It would be hugely helpful for the authors to explain the purpose of each of the metrics as they are covered in the paper. Example- What does Tajima’s D tell us? How is it different from Fu’s F?

Ln 269- The phrase “indicate diverse variation of tested populations or di-verse and results’ is unclear. Please explain.

Fig 3 legend. How was the model of population expansion calculated?

Fig 4 legend. How was the min spanning network determined?

Starting at Ln 417- This paragraph explains the presence of Wolbachia in Ae albopictus and its possible significance in vector competence. However, there was no data presented in this work about Wolbachia. Please remove this paragraph from the Discussion section. In addition, this sentence is unclear- ”Although the vector competence of distinct mosquito populations may differ 417 [59], such as Wolbachia, maternally inherited intracellular bacterial symbionts, are fre- 418 quently found in mosquitoes and are considered to be of key importance for the spread of 419 human infections”.

Ln 458- The authors state that “Previous studies have demonstrated that a combined analysis of microsatellite and 458 mitochondrial data leads to more information”. How and why is this the case? Exactly what is the added information?

Ln 466- Please briefly redefine Na and AR and explain the implications of no difference in Na and AR.

Ln 501- How does the data presented here inform vector control?

Author Response

Reviewer #1:

We appreciate the reviewer’s comments. Our point-by-point responses are provided below.

We revised the whole manuscript according to the suggestion of the reviewer.

  1. Understanding vector genetic diversity could inform risk assessment for emerging vector-borne disease outbreaks. This study explores variation of COI, ND5, and microsatellites in Ae. albopictusmosquitoes. The authors used modern population genetics metrics to characterize markers from each population. They used quite a few different types of analyses, which makes this work particularly informative compared to similar studies. Unfortunately, they neglected to explain the purpose of each type of analysis as it was introduced.

Response: A detailed explanation was added according to the reviewer’s comment, containing the purpose and differences of each type of analysis in the section of ‘Materials and Methods’ and ‘Results’.

  1. This work is organized in the traditional Results and Discussion sections. One suggestion would be to reorganize it to a combined ‘Results and Discussion’ sections, which would allow the authors to expand their explanations of each analytical method as the ms progresses.

Response: Thanks for the clear comments. we also updated Results (i.e. Figure 4) and Discussion sections, added more discussions, and limitations to our study (i.e. L. 528-532).

  1. The availability of a complete genome for this species (Palatini U, et al. Improved reference genome of the arboviral vector Aedes albopictus. Genome Biol. 2020 Aug 26;21(1):215.See also vectorbase.org) makes the use of microsatellites somewhat outdated. Understanding that short read similarity searches have limited utility, I recommend that the authors BLAST all the primers used to determine which genes were likely represented in their analyses. See https://vectorbase.org/vectorbase/app/workspace/blast/new

Response: We agree with your comments “Understanding that short read similarity searches have limited utility”. As per your recommendations, we searched BLAST with all markers, which were shown to be highly similar to the genome of Aedes albopictus. We have created an Excel document for this, so please refer to it (attached). We understand that analyzing the genome has proven to be a more effective method. Furthermore, this result is consistent with genome-based studies, so we think it's supportive research.

  1. Ln 60- The authors refer to ‘fatal’ diseases several times in the Introduction. The disease burden caused by arboviruses is much greater than case fatality rates per se. I recommend removing the word ‘fatal’ and replacing it with a word or phrase that more accurately summarizes the impact of these arboviruses on human health.

Response: We updated and replaced with synonym word "lethal." We also added an extra sentence in L. 53-55 about why transmitting diseases is dangerous.

  1. Ln 76- Mosquitoes don’t ‘cause’ disease. Rather, mosquitoes transmit pathogens (the pathogens cause disease..)

Response: Based on the reviewer's comments, we corrected the phrase corresponding to Ln 79.

  1. Ln 83- This sentence is unclear- please clarify- In foreign cases, imported cases have led to indigenous infection.

Response: Based on the reviewer's comments, we have rewritten these sentences for clarity in Ln 88-90.

  1. Ln 85- This sentence is unclear- please clarify- However, investigations of travelers having introduced the infection 85 from overseas have continuously included cases of dengue fever 194 and chikungunya 11 86 cases in 2018 only.

Response: Based on the reviewer's comments, we have rewritten these sentences for clarity in Ln 90-92.

  1. Ln 99- The authors state that their goal is to “to determine the introduction of Ae. albopictus in Korea”. Did the reported analyses satisfy that goal?

Response: Thanks for the clear comments. It's too early to make a judgment about the introduction. Therefore, it was corrected 'to understand the genetic status' of the Korean Ae. albopictus population. We updated the Figure 4 and supplementary Figures S2 and S3, where we try to make. 

  1. Ln 224- The authors state that population VTE has the largest number of haplotypes. However, this evidence is not clear in either Table 1 or Table S2. Please clarify….

Response: Already mentioned in Table, but "Results" seems to require additional explanation, so we put an extra sentence in Ln 240-242.

  1. Ln 226- Please define all acronyms upon first use, eg., VL and TJ.

Response: According to the reviewer's comment, we wrote the full name when we first mentioned it.

  1. Table 2 has a typo.

Response: It was checked and fixed.

Table 2. Some readers may not understand the differences between ‘population and group-wise comparisons. Please add a sentence to the Results section, explaining the differences between the metrics.

Response: Thanks for the reviewer's comment. we put additional sentences in Ln 276-279 to make it easier for readers to understand.

  1. Starting in Ln 235- It would be hugely helpful for the authors to explain the purpose of each of the metrics as they are covered in the paper. Example- What does Tajima’s D tell us? How is it different from Fu’s F?

Response: Added text about why we have a neutral test and its meaning (Ln 289-291). we also explained the differences between Fu's F and the Tajima test on Ln 291-293.

  1. Ln 269- The phrase “indicate diverse variation of tested populations or di-verse and results’ is unclear. Please explain.

Response: Corrected unclear sentences to Ln 293-295 based on the reviewer's comments.

  1. Fig 3 legend. How was the model of population expansion calculated?

Response: We've added an extra sentence to the Figure 3 legend about how this was analyzed. We also mentioned Ln 197-198 in "Materials and Methods."

  1. Fig 4 legend. How was the min spanning network determined?

Response: We've added an extra sentence to the Figure 4 legend about how to make this haplotype network. And I also mentioned Ln 199-203 in "Materials and Methods."

  1. Starting at Ln 417- This paragraph explains the presence of Wolbachia in Ae albopictus and its possible significance in vector competence. However, there was no data presented in this work about Wolbachia. Please remove this paragraph from the Discussion section. In addition, this sentence is unclear- ”Although the vector competence of distinct mosquito populations may differ 417 [59], such as Wolbachia, maternally inherited intracellular bacterial symbionts, are fre- 418 quently found in mosquitoes and are considered to be of key importance for the spread of 419 human infections”.

Response: Thanks for the clear comments. we agree with reviewer comment as we don’t have any supported data on Wolbachia infection. So, we have minimized and fixed the issue (Starting at Ln 458).

  1. Ln 458- The authors state that “Previous studies have demonstrated that a combined analysis of microsatellite and 458 mitochondrial data leads to more information”. How and why is this the case? Exactly what is the added information?

Response: Just mitochondrial DNA (mt) could not express itself clearly, but when mt and microsatellites were combined, greater resolutions were achieved. Microsatellites are commonly utilized in previous studies (see references 79–81) to evaluate mosquitoes' genetic variety and structure (see reference 82). In short, we explained why we employed microsatellites as a marker (Ln 488-494).

  1. Ln 466- Please briefly redefine Na and AR and explain the implications of no difference in Na and AR.

Response: We have updated the abbreviation of NA and AR and meaning of them for more clarification (Ln 499-502).

  1. Ln 501- How does the data presented here inform vector control?

Response: Based on the reviewer's advice, we added the sentences in discussion section (Ln 541-544), and conclusion section (Ln 544-558).

Reviewer 2 Report

The manuscript “Genetic Diversity of Dengue Vector Aedes albopictus collected from South Korea, Japan, and Laos” by Shin et al. used 16 microsatellite markers in addition to the two mt markers to characterize the genetic variability in Aedes albopictus. There are serious flaws in the manuscript that prevent me to recommend its publication. 

The manuscript would benefit from professional copyediting. There are many awkward sentences, typos, and inconsistencies throughout the manuscript. 

The sampling effort for the microsatellite analyses is inadequate. It is widely accepted that 30 specimens are needed per population to achieve enough statistical power to account for the natural variance found within populations to reach reliable conclusions. In this study, the authors have not included more than 12 specimens per population, with values as low as 2 for 3 populations. This represents a major limitation of the study. 

There is no information on how the authors planned to assess Aedes albopictus introduction pathways. None of the analyses presented in the manuscript are adequate to address this hypothesis. Furthermore, their conclusions are not supported by the results as in lines 462-463: “In this study, microsatellite analysis showed that some Korean populations originated from other countries while others originated in Korea.” 

Author Response

Reviewer #2:

We appreciate the reviewer’s comments. Our point-by-point responses are provided below.

We changed this sentences according to the suggestion of the reviewer.

The manuscript “Genetic Diversity of Dengue Vector Aedes albopictus collected from South Korea, Japan, and Laos” by Shin et al. used 16 microsatellite markers in addition to the two mt markers to characterize the genetic variability in Aedes albopictus. There are serious flaws in the manuscript that prevent me to recommend its publication. 

 Response: Thanks for your valuable observation and suggestions. We updated the whole manuscript as you recommend and other reviewers’ comments.

The manuscript would benefit from professional copyediting. There are many awkward sentences, typos, and inconsistencies throughout the manuscript. 

Response: Thanks for the good comments. We performed English proofreading according to the reviewers' comments (attached)

The sampling effort for the microsatellite analyses is inadequate. It is widely accepted that 30 specimens are needed per population to achieve enough statistical power to account for the natural variance found within populations to reach reliable conclusions. In this study, the authors have not included more than 12 specimens per population, with values as low as 2 for 3 populations. This represents a major limitation of the study. 

Response: We mentioned our limitations in discussion section (Ln 528-532) that the number of samples used for microsatellite analysis should be increased. Therefore, we focused on the analysis of mitochondrial genes in this study. Microsatellite analysis was used to support the results of mitochondrial genetic study. According to mitochondrial gene analysis, the Korean population shared haplotypes with other countries, and microsatellite analysis also showed similar results.

There is no information on how the authors planned to assess Aedes albopictus introduction pathways. None of the analyses presented in the manuscript are adequate to address this hypothesis. Furthermore, their conclusions are not supported by the results as in lines 462-463: “In this study, microsatellite analysis showed that some Korean populations originated from other countries while others originated in Korea.” 

Response:

Thanks for the reviewer's comment. As noted by the reviewer, we did not analyze enough data to support this hypothesis. However, to support this to some extent, we performed additional data analysis (Figure 4 and Supplementary Figures S2 and S3) and confirmed the haplotype network by combining the data generated with this manuscript and the data from the NCBI download, which provided meaningful findings. According to findings from previous research, some Korean populations shared haplotypes with other countries, but some shared only Korean clades (see the reference no. 33) 

The reference to introduction pathways is large relative to the data, so it has been removed and summarized in a separate sentence as the reviewer's opinion (Ln 495-496).

Author Response

Reviewer #3:

We appreciate the reviewer’s comments. Our point-by-point responses are provided below.

We have changed sentences according to the suggestion of the reviewer.

The article of J. Shin with coauthors explores population structure of Aedes albopictus mosquitoes in South Korea using 2 mitochondrial genes and 16 microsatellites. They conducted standard population genetic analyses based on 19 populations in Korea and two populations from the neighboring countries – Japan and Laos. The authors concluded that the population structure of Ae. albopictus is a result of a mixture of the native and recently introduced mosquitoes. Although the study is interesting and significant, I suggest a major revision of the article to make the results more conclusive. Below are my suggestions for the article improvement.

Major comments:

  1. Because the authors used only two populations outside of Korea, their results about the origin of the invasive component of the Ae. albopictus mosquitoes in this country is very inconclusive. I suggest to use the data for mitochondrial genes that are available in the GenBank and other databases to include more samples from the neighboring countries and reanalyze the data.

Response: Based on the reviewers' opinions, we performed a re-analysis using mitochondrial genetic data from several countries, including South Korea, from NCBI. A haplotype network that analyzed the NCBI data and the data in this paper showed that some Korean populations shared haplotypes with other countries. The analysis yielded significant results, clearly supporting our hypothesis. The results of this analysis have been added to Ln338-343 in the text, Figures 4, Figure S2, and S3 in the supplementary data.

  1. I also suggest to perform hierarchical cluster analyses (R_Core_Team, 2021) that may provide more conclusive results about clusters of Ae. albopictus mosquitoes inside Korea.

Response: We agree with the reviewer’s excellent suggestions and comments. We think the analysis mentioned by the reviewer is also helpful in this paper. The reviewer's analysis will be used in later research to analyze more data, such as Exom sequencing. And we have planned to investigate using additional Korean populations with more individuals.

  1. I also think that the authors need to subdivide populations inside Korea based on their geographical origin into, for example, south costal, central, and northern areas. It would be a good idea to indicate populations from these three areas by different colors on the figures to see whether or not any geographical component is involved in the population structure in Korea.

Response: We performed an analysis based on the reviewers' comments as subdivided populations inside Korea based on their geographical origin. According to an AMOVA analysis based on mitochondrial DNA, the Korean population had non-significant findings in group conditions such as east-west, north-south, and provincial-based. We mentioned the results in supplementary data (Table S4).

  1. It is unclear why different analyses were performed for the microsatellite and mitochondrial genes. Please, unify your research or explain why you conduct your analyses this way.

Response: We have updated the whole manuscript as per your comments and the other reviewers’ comments. In this study, the results/outcomes of microsatellite and mitochondrial genes are quite similar, but the analysis programs are different. There is also a point that the properties of both markers are complemented each other. Microsatellite data are loci-based, whereas mitochondrial data are sequence-based. This variation might influence how the analytic programs are used. In this study, we intend to support the hypothesis with both data (mt-DNA and microsatellites) that support the same outcome.

Minor comments:

  1. In the abstract (line 35) the authors wrote: “Based on the findings, two hypotheses can be proposed. First, these species have existed in Korea as native species. Second, it evolved into the Korean meta population following one or more invasions from other south-east Asian countries, including Laos and Japan”. Please, clarify if these are two alternative hypotheses or if these two hypotheses are not mutually exclusive.

Response: We revised the abstract and updated the unclear abstraction based on the reviewer's opinion.

  1. In line 358 the authors wrote: “The two clusters did not significantly differ from one another. At the highest value (K= 4), Korean populations had distinct genetic groups, and there was no unique genetic group in Korean populations.” The meaning of these sentences is very hard to understand, please explain more clearly what does it mean.

Response: We modified and re-phrased the sentences. We corrected the sentence to be more precise, and there is a corrected sentence on Ln 398-402.

  1. I made some minor text corrections that need to be addressed. Please, see attached manuscript text with corrections.

Response: Thanks for the reviewer's text corrections. All the documents, including supplements, have been corrected.

  1. I also suggest to do an additional English correction that is available and is of a good quality at MDPI.

Response: Following the reviewers' comments, the manuscript underwent complete English editing after revision (attached certificate).

Round 2

Reviewer 1 Report

The authors have taken steps to substantially improve the clarity of the article in their revisions.

Define which are ‘within group’ and which are “within population”

Lns 361-362- These back-to-back statements are redundant. Please clarify ”Haplotypes H-17 and H-18 originated from the Japanese population (H21). Therefore, these Korean populations may have orig- inated from the Japanese population.”

Table 2- is missing the ‘p’ in populations.

Discussion

Lns ~501-506, the authors state that “The Korean population of Ae. albopictus has a distinct genetic structure within the Korean population that was not observed in the mt analysis,(COI and ND5) analyses. However, this genetic structure was not notably different from that of the other populations in Laos and Japan (Table 4, Fig 5)”. However, the clades in Figs 4 and S2 were determined based on COI sequences. Please justify your statement.

>>> where is Table 4?

Lns 510-517. The authors conclude that Korean populations showed low mt genetic diversity. However, sample sizes were very low in some locations. Please add a statement to the Discussion stating this as a limitation to drawing strong conclusions in this regard.

Lns 597- 599- Specifically which genetic features are consistent with Korea being the original source of Ae. albopictus?

Ln 599- change ’Korean’ to Korea’

Ln 606- The sentence stating “The introduction of the Ae. albopictus mosquito is further supported by genetic diversity…” is unclear. The Introduction to what location, specifically?

There are several typos of “Ae. albopticus..”

In future work, please add the genomic locus information to supplement the microsatellite data. Sanger sequencing of amplicons would be an important validation step in this regard. This would substantially improve the biological relevance of the conclusions drawn.

Author Response

Reviewer 1

Comments and Suggestions for Authors

The authors have taken steps to substantially improve the clarity of the article in their revisions.

Define which are ‘within group’ and which are “within population”

Response: To avoid confusion, we modify sentences in Ln 176-178 to define AMOVA’s Source of variation.

Lns 361-362- These back-to-back statements are redundant. Please clarify”Haplotypes H-17 and H-18 originated from the Japanese population (H21). Therefore, these Korean populations may have orig- inated from the Japanese population.”

Response: We make these sentences clear on Ln 322-324

Table 2- is missing the ‘p’ in populations.

Response: We added the ‘p’ in Table 2’s second cell.

Discussion

Lns ~501-506, the authors state that “The Korean population of Ae. albopictus has a distinct genetic structure within the Korean population that was not observed in the mt analysis, (COI and ND5) analyses. However, this genetic structure was not notably different from that of the other populations in Laos and Japan (Table 4, Fig 5)”. However, the clades in Figs 4 and S2 were determined based on COI sequences. Please justify your statement.

>>> where is Table 4?

Response: We described microsatellite results. So, we checked the data file and corrected the text (Table 4 to Table S5, Fig 5 to Ln 508).

In addition, we added more discussion line no (Ln 436-448) based on Figs 4 and S2, which were determined based on mitochondrial analysis (COI and ND5 sequences).

Lns 510-517. The authors conclude that Korean populations showed low mt genetic diversity. However, sample sizes were very low in some locations. Please add a statement to the Discussion stating this as a limitation to drawing strong conclusions in this regard.

Response: We have already mentioned the limitations of this study related to low sample sizes in Ln 459-461, and the conclusion as well as further limitations were included in Ln 538-546.

Ln no’s 597- 599- Specifically which genetic features are consistent with Korea being the original source of Ae. albopictus?

Response: We mentioned the characteristics and genetic features (the features mention in Ln 528-531, genetic features mention in Ln. 488-490, 511-513)

Ln 599- change ’Korean’ to Korea’

Response: according to the reviewer’s comment, we changed ‘Korean’ to ‘Korea’.

Ln 606- The sentence stating “The introduction of the Ae. albopictus mosquito is further supported by genetic diversity…” is unclear. The Introduction to what location, specifically?

Response: To avoid confusion, we put the location name Ln 539-542 (The introduction of the ‘Korea’ Ae. albopictus….).)

There are several typos of “Ae. albopticus..”

 Response: We modified every typo ‘Ae. albopticus’ to ‘Ae. albopictus’.

In future work, please add the genomic locus information to supplement the microsatellite data. Sanger sequencing of amplicons would be an important validation step in this regard. This would substantially improve the biological relevance of the conclusions drawn.

Response: We agree with your comments. So, we mentioned our limitation on Ln 539-542, making a supplement of the BLAST report using Vectorbase (Table S8), and mentioned Materials and Methods on Ln 197-198.

Reviewer 2 Report

-

Author Response

Reviewer 2

Comments and Suggestions for Authors

No comments.

Response: Thanks for your valuable efforts, comments, and suggestions. We tried to follow the majority suggestions of all respective reviewers’ recommendations and their valuable suggestions. For the improvement our manuscript, we made considerable revisions, English proofreading, and significant changes to the manuscript. We are aware of this study's limitations. Please understand of our limitations in this present manuscript, though we have modified it as much as possible to overcome this issue. In the future study, we'll conduct further study and collect more samples for a new publication.

Reviewer 3 Report

I think that authors significantly revised the manuscript and provided detailed responses to my comments and suggestions. They also significantly improved their original text by editing and correcting their language. Although they did not address all the comments that I suggested I think that the manuscript can be published in its current form.

Author Response

Reviewer 3

Comments and Suggestions for Authors

I think that authors significantly revised the manuscript and provided detailed responses to my comments and suggestions. They also significantly improved their original text by editing and correcting their language. Although they did not address all the comments that I suggested I think that the manuscript can be published in its current form.

Response: We are very grateful to your valuable time, thoughtful comments, and kind opinions. Your earlier suggestions that helped to improve this manuscript a lot.

In the revised version of the present manuscript, we added content, English correction, and final proofread version of MS. Thanks to the reviewer again.
